# Underactuated Humanoid Peeling Approach for Pickled Mustard Tuber Based on Metamorphic Constraints

**DOI:** 10.3390/biomimetics8080566

**Published:** 2023-11-24

**Authors:** Haochuan Wan, Lei Chen, Jiayu Xiao, Nana Chen, Hankun Yin, Lin Zhang

**Affiliations:** 1School of Robotics Engineering, Yangtze Normal University, Chongqing 408100, China; 20180002@yznu.edu.cn (H.W.); 202124591120@stu.yznu.edu.cn (J.X.); linzhang-sjtu@sjtu.edu.cn (L.Z.); 2School of Artificial Intelligence, Anhui University of Science and Technology, Huainan 232001, China; 3School of Intelligent Manufacturing Engineering, Chongqing University of Arts and Sciences, Chongqing 402160, China; yinhankun@cqwu.edu.cn

**Keywords:** pickled mustard tuber, metamorphic constraints, peeling and de-stringing, bio-inspired robotics, end effector trajectory

## Abstract

Pickled mustard tuber (PMT), also known as *Brassica juncea* var. *tumida*, is a conical tuberous vegetable with a scaly upper part and a coarse fiber skin covering the lower part. Due to its highly distorted and complex heterogeneous fiber network structure, traditional manual labor is still used for peeling and removing fibers from pickled mustard tuber, as there is currently no effective, fully automated method or equipment available. In this study, we designed an underactuated humanoid pickled mustard tuber peeling robot based on variable configuration constraints that emulate the human “insert-clamp-tear” process via probabilistic statistical design. Based on actual pickled mustard tuber morphological cluster analysis and statistical features, we constructed three different types of pickled mustard tuber peeling tool spectral profiles and analyzed the modular mechanical properties of three different tool configurations to optimize the variable configuration constraint effect and improve the robot’s end effector trajectory. Finally, an ADAMS virtual prototype model of the pickled mustard tuber peeling robot was established, and simulation analysis of the “insert-clamp-tear” process was performed based on the three pickled mustard tuber statistical classification selection. The results showed that the pickled mustard tuber peeling robot had a meat loss rate of no more than 15% for each corresponding category of pickled mustard tuber, a theoretical peeling rate of up to 15 pieces per minute, and an average residual rate of only about 2% for old fibers. Based on reasonable meat loss, the efficiency of peeling was greatly improved, which laid the theoretical foundation for fully automated pickled mustard tuber peeling.

## 1. Introduction

Fermented vegetables are increasingly gaining attention due to their rich nutritional value and health benefits for consumers. *Brassica juncea* var. *tumida* Tsen & Lee, also known as pickled mustard tuber, is an important agricultural and economic crop in China, especially in the Yangtze River region. Pickled mustard tuber is commonly consumed in salted or pickled form and is referred to as “Zha Cai” in China [1]. It is a conical tuberous vegetable with scale-like upper parts and fibrous skin wrapping the lower parts. It is considered one of the world’s top three famous pickled vegetables [2], possibly due to its high content of essential nutrients such as vitamin C, vitamin K, various B vitamins, and dietary fiber that promote digestion and reduce constipation. Additionally, pickled mustard tuber is rich in important minerals such as calcium, iron, and potassium, which help maintain bone health and electrolyte balance in the body [3]. With increasing consumer demands for food quality and safety, the necessity of fully automated deep processing is evident. Automation can reduce human errors, improve product safety, flavor, production efficiency, and lower production costs. The deep processing of pickled mustard tuber includes procedures such as removing fibrous skin, pickling, trimming, washing, and jar filling. Among these, removing fibrous skin is the core process in the deep processing industry of pickled mustard tuber. However, due to the highly distorted and randomly shaped surface of pickled mustard tuber, as well as the difficulty in perceiving the heterogeneous fibrous structure inside, effective methods for removing fibrous skin have not been found in current research. On the other hand, the complex technical requirements for measuring the random and highly distorted shape of pickled mustard tuber, the unknown location of the heterogeneous fibrous structure inside, and the extremely low-cost equipment required for removing fibrous skin pose challenges and limitations to the progress of current research in this field. Therefore, research on methods and equipment for removing fibrous skin from pickled mustard tuber is urgently needed.

There is relatively more research on the removal methods of tough fibers and skin in fruits and vegetables with relatively regular surface morphology and simpler internal fibrous structures [4,5,6]. Common methods for removing tough fibers and skin in fruits and vegetables include infrared skin–flesh separation, ultrasonic vibration removal, alkaline corrosion peeling, and centrifugal separation. Infrared skin–flesh separation technology utilizes infrared radiation to quickly heat up the fruit’s surface, followed by cold water cooling to achieve the purpose of separating the skin from the flesh while preserving the nutritional components of the fruit [7]. The ultrasonic vibration removal method utilizes the vibrational effect of ultrasound to soften the tough fibers in fruits and vegetables, making them easier to remove [8]. Alkaline corrosion peeling technology utilizes the corrosive properties of the alkaline solution to dissolve the adhesive layer on the fruit’s surface, thus separating the peel [9]. The centrifugal separation removal technique, based on the principle of centrifugal force [10], can separate the tough fibers in fruits and vegetables, thereby removing them.

However, the current process of removing fibers and peeling the skin of pickled mustard tuber still relies on traditional manual methods. In response to this, Xie et al. proposed a new type of circular device for removing fibers and peeling pickled mustard tuber [11]. This device follows the basic process of mimicking manual peeling, which involves inserting a knife into the pickled mustard tuber, using the knife and fingers to grip the skin, and tearing off the skin along the cutting line of the vegetable. However, this method always uses a fixed motion trajectory, which has low applicability to pickled mustard tubers of different shapes. It not only increases the loss of meat but also reduces the level of automation in the peeling process of pickled mustard tuber. These results in issues such as high labor intensity, high costs, and low efficiency, limiting the development of deep processing and automated production of pickled mustard tuber. To address the problems associated with fixed cutting trajectories in peeling fruits and vegetables, scholars have proposed flexible peeling methods and devices [12,13,14,15], which have been applied in the peeling operations of garlic [16], lychee [17], lotus root [18], and other fruits and vegetables. These methods have improved the universality of flexible operations while maintaining a lower rate of meat damage and improving peeling efficiency. Compared to manual labor, these methods can increase work efficiency by more than 20 times, reduce labor intensity, and achieve high efficiency in peeling while preserving the integrity and original taste of the final product.

This article addresses the challenges posed by the highly distorted and heterogeneous structure of pickled vegetable skin, as well as the random distribution of fibrous structures. Drawing inspiration from the dexterity, adaptability [19], multimodal perception capabilities [20], and synergistic effects of human hands and fingers [21], a human-like pickling robot based on the “insert-clamp-tear” process is designed with underconstrained architecture constraints. A modular force analysis is conducted using variable configuration constraints to optimize different flexible deformation forces based on the clustering analysis of different pickled vegetable shapes. Based on probability statistics, the “insert-clamp-tear” process of human-like peeling is simulated for different categories of pickled vegetables. The feasibility of the statistical analysis and design results is verified via virtual prototype simulations using Adams 2022. This greatly improves the efficiency of peeling while maintaining reasonable flesh loss, providing a theoretical and technical basis for fully automated “fiber-removing” of pickled vegetables.

## 2. Design of Pickled Mustard Tuber Peeling Robot

### 2.1. Human-Like Peeling Process

The manual peeling method is shown in Figure 1a. First, the cutting tool is inserted into the skin and flesh of the pickled mustard tuber. Then, the thumb and the knife work together to clamp the fibrous skin of the tuber. Finally, the knife and fingers work together to tear off the fibrous skin along the line of the pickled mustard tuber. The overall process flow of the human-like peeling process of the pickled mustard tuber peeling robot is shown in Figure 1b. It adopts a primary–secondary interaction mechanism that can better adapt to the shape of the pickled mustard tuber. Based on the “insert-clamp-tear” human-like peeling process simulation, three different compliant peeling behaviors can be generated by a single motor control input for the peeling mechanism. In the first “insertion” mode, the electric push rod moves downward, and the upper peeling claw inserts between the flesh and fibrous skin of the pickled mustard tuber. In the second “clamping” mode, the upper and lower peeling claws and the skin of the pickled mustard tuber will form a clamping limit. The third “tearing” mode is achieved by the upper and lower peeling claws clamping the fibrous skin of the pickled mustard tuber while expanding outward. This causes the upper peeling claw to move downward and extend outward, with the lower peeling claw tearing off the fibrous skin of the pickled mustard tuber. Finally, the bottom motor rotates to complete the peeling process and remove the fibrous skin of the pickled mustard tuber [22].

### 2.2. Metamorphic Mechanism Simulation

The pickled mustard tuber peeling mechanism is shown in Figure 2, which mainly consists of an electric push rod, a positioning claw, a peeling claw, a connecting rod, a spring system, and a motor. The spring system consists of four sets, each consisting of two tension springs and three compression springs. The tension springs are used to cut the pickled mustard tuber, while the compression springs are used to maintain the relative position of the upper and lower peeling claws [23]. The proposed pickled mustard tuber peeling mechanism has three degrees of freedom, namely X, Y, and theta [24]. During the cutting tool configuration transformation process, the electric push rod moves downward, driving the connecting rod and the cutting tool. The connecting rod and the cutting tool remain relatively stationary, and the RRRP rod group degenerates into an RRP rod group, where “R” represents rotary pairs and “P” represents prismatic pairs in the rod group. When the cutting tool is inserted into the pickled mustard tuber, it is moved toward the center by the spring tension and cuts the skin of the pickled mustard tuber. When the cutting tool contacts the lower peeling claw, the upper and lower peeling claws and the fibrous skin of PMT form a clamping limit. Subsequently, the RRRP rod group changes into an RRR rod group. Finally, under the movement of the upper and lower peeling claws expanding outward, the fibrous skin of the pickled mustard tuber is torn off. The RRRP rod group changes back into an RRP. Based on kinematics, the cutting tool (pickled mustard tuber peeling mechanism) uses three variable configurations to complete the peeling of pickled mustard tuber with significant surface distortion and morphological characteristics.

### 2.3. System Architecture of PMT Peeling Approach

The integrated design of the pickled mustard tuber peeling robot follows a research framework consisting of perception and localization, fibrous structure analysis, construction of cutting trajectory, execution of peeling, adaptive adjustment, and data recording and analysis. Firstly, a force measuring mechanism is used to obtain the force curve of the pickled mustard tubers, and based on the peak force distribution, the distribution of fibrous structures on the surface is determined to establish an appropriate cutting trajectory. The upper and lower peeling claws are then inserted according to the position and quantity of the fibrous structures. Via the internal sliding of the guiding and adjusting grooves, the claws are adjusted to the desired position. The pickled mustard tuber is positioned using a central block and a placement claw plate, with the positioning claws inserted to secure it. The upper peeling claws penetrate the tuber with fibrous structures, and the piston cylinder control leads to the downward movement to form clamping between the upper and lower peeling claws. As the piston cylinder descends, the upper peeling claws gradually unfold and adjust their angles along with the connected rod and lower peeling claws, driven by the derived force and gradually expanding outward. As the lower peeling claws unfold, the derived upper peeling claws cooperate with an auxiliary limit wheel for angle adjustment, resulting in synchronous displacement and the tearing of the fibrous skin. Finally, the fibrous meat with the fibrous structures is removed by the rotation of the bottom motor. After peeling, the first adjustment disc rises, the second spring resets to retrieve the second claw rod, the first spring resets to retrieve the first claw rod, and the internal damping of the piston damping cylinder resets. The derived transfer block then drives the auxiliary limit wheel to reset the outer peeling claws.

## 3. Mechanical Analysis of Metamorphic Peeling Finger

Based on the composition principle of a constraint–variable-cell mechanism using an extended Assur linkage assembly [25], mathematical relationships are established between kinematical and mechanical analysis models of each basic unit, including active components, Assur linkage assemblies, and extended Assur linkage assemblies. In the variable cell process, each cell pair is constrained sequentially according to the required working stage, and the constrained extended Assur linkage assembly degenerates into the corresponding equivalent Assur linkage assembly. Therefore, kinematical and mechanical analysis of the extended Assur linkage assembly can be conducted according to different configurations, with analysis of the corresponding equivalent Assur linkage assembly performed for each configuration [26]. The muscle-stripping mechanism studied in this paper belongs to the RRRP, which is one of nine extended second-level Assur linkage assemblies [27].

### 3.1. Degradation Process of Metamorphic Peeling Finger

During the configuration transformation process, the equivalent Assur linkage assembly of the RRRP extended Assur linkage assembly is shown in Figure 3. In this figure, rotary pair 2 and prismatic pair 4 are the cell-pair components. In the first configuration (I), cell-pair rotary pairs 1 and 2 are constrained and remain stationary, while components 1 and 2 become a unified structure relative to the rest. The cell-pair rotary pair 3 and prismatic pair 4 are in motion, and the RRRP assembly degenerates into an RRP assembly. In the second configuration (II), the cell-pair prismatic pair 4 is constrained and transitions from motion to a static state, while cell-pair rotary pairs 1 and 2 transition from a static state to motion. The RRRP assembly degenerates into an RRR assembly. In the third configuration (III), the cell-pair rotary pair 2 transitions from motion to a static state, while components 2 and 3 remain relatively stationary. Cell-pair rotary pairs 1 and 2, along with cell-pair prismatic pair 4, are in motion, and the RRRP assembly degenerates back into an RRP assembly. The following section will perform a modular mechanical analysis of the three configurations of the RRRP linkage assembly.

### 3.2. Modular Mechanical Analysis of Metamorphic Peeling Finger

The mechanical analysis of the three configurations is shown in Figure 4, wherein the first configuration (I), the RRRP linkage assembly degenerates into an RRP linkage assembly. Assuming there is an external force *F*_7_ and torque *T*_3_ applied to the slider, and links *N*_2_*N*_3_ and *N*_3_*N*_4_ are subjected to forces and torques due to the sliding motion of the slider, denoted as *F*_5_, *F*_6_, *T*_1_, and *T*_2_, we will analyze and explain the supporting and reaction forces on each rotary pair and the driving and constraint torques on the cell-pair rotary pair. The specific symbols are defined as shown in Table 1.

The specific steps of modular mechanical analysis are shown as follows:(1)$$\left\{\begin{array}{l} \cos\alpha =\frac{l_{14}^{2}+l_{34}^{2}-l_{13}^{2}}{2l_{14}l_{34}}=\frac{d_{h}^{2}+d_{m}^{2}-d_{k}^{2}}{2d_{h}d_{m}}\\ d_{k}=\sqrt{d_{h}^{2}+d_{m}^{2}-\frac{d_{h}d_{m}\left(l_{14}^{2}+l_{34}^{2}-l_{13}^{2}\right)}{l_{14}l_{34}}}\\ F_{k}=k\left(a-d_{k}\right)\\ F_{k}\sin\theta =\left(m_{23}g+m_{43}g\right) \end{array}\right.$$
(2)$$\left\{\begin{array}{l} \cos\theta =\frac{d_{k}^{2}+d_{m}^{2}-d_{h}^{2}}{2d_{k}d_{m}}\\ \theta =\arccos\left[\frac{2d_{m}^{2}-\frac{d_{k}d_{m}\left(l_{14}^{2}+l_{34}^{2}-l_{13}^{2}\right)}{l_{14}l_{34}}}{2\sqrt{d_{k}^{2}+d_{m}^{2}-\frac{d_{h}d_{m}\left(l_{14}^{2}+l_{34}^{2}-l_{13}^{2}\right)}{l_{14}l_{34}}}d_{m}}\right]\\ T_{k}=F_{k}d_{m}\sin\theta \end{array}\right.$$
(3)$$\left\{\begin{array}{l} R_{4x}=\frac{d_{h}^{2}+d_{m}^{2}-d_{k}^{2}}{P_{24y}}\\ R_{4y}=\frac{\left(-A_{1}+B_{1}P_{24x}\right)}{P_{24x}}\\ A_{1}=-\left(T_{54}+T_{64}+T_{1}+T_{2}\right)\\ B_{1}=-\left(F_{5y}+F_{6y}+F_{7y}\right) \end{array}\right.$$
(4)$$\left\{\begin{array}{c} R_{2x}=-\left(R_{4x}+F_{6x}+F_{7x}\right)\\ R_{2y}=-\left(R_{4y}+F_{6y}+F_{7y}\right) \end{array}\right.$$
(5)$$\left\{\begin{array}{l} T_{24}=-P_{24x}R_{2y}+P_{24y}R_{2x}\\ T_{54}=P_{54x}m_{23}g\\ T_{64}=P_{64x}m_{43}g\\ \Delta T=-T_{24}+T_{54}+T_{64} \end{array}\right.$$

In the second configuration (II), the RRRP linkage assembly degenerates into an RRR linkage assembly, and the force analysis is shown in Figure 4.

The specific steps of modular mechanical analysis are shown as follows:(6)$$\left\{\begin{array}{l} \cos\alpha =\frac{l_{23}^{2}+l_{34}^{2}-l_{24}^{2}}{2l_{23}l_{34}}=\frac{d_{1}^{2}+d_{2}^{2}-d_{k1}^{2}}{2d_{1}d_{2}}\\ d_{k1}=\sqrt{d_{1}^{2}+d_{2}^{2}-\frac{d_{1}d_{2}\left(l_{23}^{2}+l_{34}^{2}-l_{24}^{2}\right)}{l_{23}l_{34}}}\\ F_{k1}=k_{1}\left(a-d_{k1}\right) \end{array}\right.$$
(7)$$\left\{\begin{array}{l} R_{4x}=\frac{\left(A_{2}P_{43x}-B_{2}P_{42x}\right)}{P_{43x}P_{42y}-P_{43y}P_{42x}}\\ R_{4y}=\frac{\left(A_{2}P_{43y}-B_{2}P_{42x}\right)}{P_{43x}P_{42y}-P_{43y}P_{42x}}\\ A_{2}=-\left(T_{51}+T_{61}+T_{1}+T_{2}\right)\\ B_{2}=-\left(T_{63}+T_{2}+T_{k}\right) \end{array}\right.$$
(8)$$\left\{\begin{array}{c} R_{2x}=-\left(R_{4x}+F_{5x}+F_{6x}\right)\\ R_{2y}=-\left(R_{4y}+F_{5y}+F_{6y}\right) \end{array}\right.$$
(9)$$\left\{\begin{array}{c} R_{3x}=-\left(R_{4x}+F_{6x}+F_{k1x}\right)\\ R_{3y}=-\left(R_{4y}+F_{6y}+F_{k1y}\right)\\ F_{k1x}=F_{k1}\cos\left(\gamma +\theta -\pi \right)\\ F_{k1y}=F_{k1}\sin\left(\gamma +\theta -\pi \right) \end{array}\right.$$
(10)$$\left\{\begin{array}{l} T_{23}=-P_{24x}R_{2y}+P_{24y}R_{2x}\\ T_{43}=-P_{43x}R_{4y}+P_{43y}R_{4x}\\ T_{53}=P_{53x}m_{23}g\\ T_{63}=P_{63x}m_{43}g \end{array}\right.$$
(11)$$\Delta T=T_{23}-T_{43}+T_{53}-T_{63}$$

In the third configuration (III), the RRRP linkage assembly changes to an RRP linkage assembly. The calculation method for the driving torque Δ*T* of the cell-pair rotary pair and the meanings of other variables can be referred to the force analysis and calculation model of the first configuration (I) and the second configuration (II).

## 4. Trajectory Predicting of Metamorphic Peeling Finger

### 4.1. Morphological Statistical Analysis of PMT

The shape index of pickled mustard greens is an important visual quality indicator. As shown in the table, the shape index gradually decreases from 1.32 to 1.15 as the nitrogen application rate increases. Increased nitrogen application leads to an increase in the longitudinal diameter of the stem tubers [28], indicating that an appropriate nitrogen application rate can effectively improve the occurrence of rod-like shapes in the greens. This is beneficial for the cutting process of the peeling mechanism and can improve the processing yield of pickled mustard greens. Moreover, the longitudinal dimension of pickled mustard greens mainly falls between 10 cm and 20 cm. By referring to Table 2 or conducting further research, the aspect ratio of pickled mustard greens can be obtained.

As nitrogen application rates increase, various indicators of pickled mustard tuber, including plant height, plant width, vegetable shape index, skin elasticity rate, and net yield, show some differences among different codes. As shown in Figure 5. Using SPSS 2019 data processing software, a cluster analysis was conducted on the non-dimensionalized representative indicators to classify seven different types of pickled mustard tuber into three distinct categories. The first category, represented by codes *N*_600_, *N*_750_, and *N*_900_, exhibited shorter plant heights. The second category, represented by codes *N*_0_ and *N*_150_, displayed wider plant widths. The third category, represented by codes *N*_300_ and *N*_450_, showed higher net yield rates. In order to improve the peeling effect, the curved surface shape of pickled mustard greens was referenced, and 100 randomly selected pickled mustard greens were processed by binarization to extract the contour of their curved surface shape [29].

The severe distortion of randomly shaped surfaces can be seen as multiple arcs connected together. Selecting the most concentrated arc among the contour curves of 100 curved surfaces serves as the basis for designing the tool profile. Via statistical analysis and clustering analysis, it is demonstrated that a small number of discrete tools can accomplish the cutting of over 98% of pickled mustard greens. By discretizing the continuous operational requirements, the design complexity of the equipment itself is simplified while ensuring the quality of peeling (meat loss rate, peeling efficiency, cutting rate, etc.). Considering the average diameter of the upper, middle, and lower parts of pickled mustard greens, as well as the typical thickness of the fibrous skin, three interchangeable tool profiles with different diameters were designed for peeling pickled mustard greens of different shapes.

### 4.2. Configuration of Peeling Finger

Due to the complex connection structure between the holding plate and the external stripping claw, improper handling of specific internal dimensions can result in excessive local stress and severe destabilization of the mechanism. This, in turn, leads to the failure of the skinning simulation. Here, we analyze the effects of four cutting methods on the terminal trajectory, as shown in Figure 6.

(1) Single claw and single blade cutting method with bottom peeling claw fixed: Due to the limited rotation ability of the bottom peeling claw, which can only rotate around a point in the placement board and lacks the ability to move in other directions, the cutting path is restricted. Therefore, the end trajectory is relatively limited and cannot flexibly adapt to the shape changes in the pickled mustard tuber. The tool will realize reciprocating cutting by swinging back and forth, increasing the meat loss rate of pickled mustard tuber.

(2) Normal single claw and single blade cutting method: Compared with the fixed bottom peeling claw method, the normal single claw and single blade cutting method has a certain X–Y direction movement ability, which can bring a more flexible cutting path and better adapt to the shape changes in the pickled mustard tuber. Due to the asymmetric force distribution on the pickled mustard tuber, the stability of cutting is easily affected, and the end trajectory of the tool will also realize reciprocating cutting like the bottom peeling claw fixed cutting method, increasing the meat loss rate of the pickled mustard tuber.

(3) Single-arm multi-blade cutting method: Compared with the multi-arm cutting method, the single-arm multi-blade cutting method is relatively weak in adapting to different shapes of pickled mustard tuber. There is a direct structural connection between each locating claw, which will cause the resistance during cutting to be interrelated, affecting the stability of the end trajectory, making several blade end cutting trajectories the same, increasing the meat loss rate and residual skin tendon rate of the pickled mustard tuber.

(4) Multi-arm multi-blade cutting method: The multi-arm cutting method has good cutting performance and high cutting efficiency. Its symmetry and the characteristics of adding electric push rods and locating claws can make the cutting path more stable and accurate. The relative independence between each locating claw can reduce the mutual influence, and it is more adaptable to different shapes of the pickled mustard tuber. The multi-arm cutting method can produce stable and precise end trajectories. The blades are pressed against the meat of the pickled mustard tuber and cut the skin tendon downward, and the end trajectory forms a smooth curve. The cutting path of each blade is determined based on the external shape of the pickled mustard tuber.

Considering the advantages and disadvantages of the four cutting methods and the analysis of the end trajectory, the multi-arm multi-blade cutting method can provide relatively stable and precise end trajectories, better adapt to the head shape of pickled mustard tuber, and has superior cutting indicators (meat loss rate, residual skin tendon rate, cutting efficiency) compared with the other three cutting methods.

### 4.3. Peeling Trajectory Prediction

Assuming the pickled mustard tuber peeling robot has five springs with elastic coefficients *K*_1_, *K*_2_, *K*_3_, *K*_4_, and *K*_5_, and lengths *l_A_*_0_, *l_B_*_0_, *l_C_*_0_, *l_D_*_0_, and *l_E_*_0_, respectively, the peeling mechanism moves downward. We analyzed springs *A* and *B*, with spring *A* connected to the block and the pressing rod and spring *B* connected to the upper rod and the end of the connecting rod. When the block moves downward, spring *A* is compressed, and spring *B* is stretched. Let the compressed length be *l_A_*_1_. To allow the block to move downward, the driving force of the motor *M* must overcome the counteracting force of the spring *F*.

Before contact with the bottom peeling claw, when point 0 is not in contact, it is
(12)$$M\geq F=F_{1}+F_{2}$$

The forces acting on springs *A* and *B* are, respectively:(13)$$\left\{\begin{array}{l} F_{1}=K_{1}X_{1}=K_{1}\left(l_{A0}-l_{A1}\right)\\ F_{N}=K_{2}X_{2}=K_{2}\left[\frac{\left(h+l_{A0}-l_{A1}\right)l_{B0}}{h}-l_{B0}\right] \end{array}\right.$$
where *h* is the vertical distance from the upper end to the lower end of spring *B* in centimeters. The components of the forces acting on spring *B* in the *X* and *Y* directions are
(14)$$\left\{\begin{array}{l} F_{NX}=K_{2}\left[\frac{\left(h+l_{A0}-l_{A1}\right)\sqrt{l_{B0}^{2}-h^{2}}}{h}\sqrt{l_{B0}^{2}-h^{2}}\right]\\ F_{NY}=F_{N}\sin\theta =K_{2}\left(l_{A0}-l_{A1}\right) \end{array}\right.$$

Then, the driving force of the motor *M* is
(15)$$M\geq \left(K_{1}+K_{2}\right)X_{1}=\left(K_{1}+K_{2}\right)\left(l_{A0}-l_{A1}\right)$$

Note: $0\leq X_{1}\leq \frac{M}{K_{1}+K_{2}}$, the range of downward movement for the upper peeling claw is from 0 to 22 cm, but the vertical displacement before it encounters the bottom peeling claw is from 0 to 2 cm.

At this moment, the movement trajectory of point 0 at the end of the tool is a straight line.

When point 0 contacts the bottom peeling claw:

At this moment, the tool is subjected to the combined force of springs *C* and *D*. Spring *C* is connected to the second claw rod and the positioning claw at its two ends, while spring *D* is connected to the bottom peeling claw and the auxiliary positioning disc. As the tool moves downward, spring *C* is stretched, and spring *D* is compressed. Let the length of the stretched spring *C* be *l_C_*_1_ and the length of the compressed spring *D* be *l_D_*_1_.

The tension force acting on spring *C* is
(16)$$\left\{\begin{array}{l} F_{C}=K_{3}X_{3}=K_{3}\left(l_{C1}-l_{C0}\right)\\ F_{CY}=K_{3}X_{3}\sin\theta =K_{3}\sin\theta \left(l_{C1}-l_{C0}\right)\\ F_{CX}=K_{3}X_{3}\cos\theta =K_{3}\cos\theta \left(l_{C1}-l_{C0}\right) \end{array}\right.$$

The tension force acting on spring *D* is
(17)$$\left\{\begin{array}{l} F_{D}=K_{4}X_{4}=K_{3}\left(l_{D0}-l_{D1}\right)\\ F_{DY}=K_{4}X_{4}\sin\theta =K_{4}\sin\theta \left(l_{D0}-l_{D1}\right)\\ F_{DX}=K_{4}X_{4}\cos\theta =K_{4}\cos\theta \left(l_{D0}-l_{D1}\right) \end{array}\right.$$

If we neglect friction and other resistive forces, according to Newton’s second law, we can obtain the following equation when the end point of the tool, point 0, reaches position *X*.
(18)$$\left\{\begin{array}{l} F_{DX}-F_{CX}=m_{1}a_{1}\\ M-F_{DY}-F_{CY}=m_{1}a_{2} \end{array}\right.$$
where *m*_1_ represents the mass of the bottom peeling claw and represents the acceleration of the bottom peeling claw in the *X* and *Y* directions.

At this moment, the movement trajectory of point 0 at the end of the tool is a smooth curve.

## 5. Simulation and Analysis

### 5.1. Virtual Prototyping Modeling

Based on the standard spring element “Spring” provided by Adams, virtual substitute springs [30] are established for each spring. Combining the actual mechanical structural characteristics of the peeling mechanism, the Adams simulation model is simplified. The components that need to be debugged are connected to their respective positions for simulation debugging. The model includes a total of three fixed pairs, three moving pairs, and three rotating pairs. Each component in the Adams model is named according to its position and main motion characteristics, as shown in Table 3.

The dynamic cutting process is shown in Figure 7. As one of the most critical components of the peeling mechanism, the spring needs to be specially tuned and regulated during simulation to model the structural characteristics of the peeling mechanism and simulate the fixed roles of the bottom peeling claw and placing plate during the cutting and squeezing of the pickled mustard tuber. This enables the realization of an ideal cutting trajectory. The peeling mechanism has three degrees of freedom, while the independent control variable is only the pushing force of the electric push rod from top to bottom, which achieves underactuated control.

### 5.2. Peeling Trajectory Prediction

#### 5.2.1. Peeling Mechanism Optimization

In order to optimize the simulation results and improve the stability of the mechanism, the peeling mechanism is divided into three size specifications [31], each corresponding to the respective longitudinal diameter of the pickled mustard tuber. For pickled mustard tubers with a longitudinal diameter ranging from 20 cm to 17 cm, the height of the up_mid2 connecting rod is designed as 210 mm, and other components are proportionally reduced to obtain the specification “a” peeling mechanism. For pickled mustard tubers with a longitudinal diameter ranging from 16 cm to 14 cm and from 13 cm to 10 cm, the height of the up_mid2 connecting rod is designed as 165 mm and 125 mm, respectively, and other components are proportionally reduced based on the original model to obtain the specifications “b” and “c” peeling mechanisms. Ultimately, by proportionally reducing the dimensions of the components, three different specifications of the peeling mechanism are obtained to meet the peeling requirements of pickled mustard tubers with different longitudinal diameters while improving the stability of the mechanism and optimizing the simulation results.

#### 5.2.2. Peeling Trajectory Verification

In order to improve the reliability and applicability of the cutting device and ensure the stability of the cutting path, aiming to effectively remove the fibrous skin and improve the meat yield, simulation experiments are conducted for different shapes of pickled mustard tubers based on the corresponding peeling mechanism models. The simulation is optimized to obtain the parameters of the spring system under different transverse diameter ratios.

In Table 4, Table 5 and Table 6, the dimensionless unit for “Longitudinal Stem/Transverse Stem” is not specified; the unit for “Transverse Stem Size” is cm; the unit for “Stiffness Coefficient” is N/m; the unit for “Damping Coefficient” is N/(m/s); and the unit for “Pre-tightening Force” is N. The specific simulation data are as follows:

By analyzing the simulation trajectories and data in Figure 8 and Table 7, it can be concluded that the peel loss rate of each type of pickled mustard tuber does not exceed 15%, and the residual rate of fibrous skin does not exceed 10%. The theoretical peeling rate can reach 15 pieces/min. Based on reasonable meat loss, the peeling efficiency is greatly improved. When calculating the volume of the cutting trajectory during the restoration process, there may be a large error between some fitting functions and the actual cutting trajectory. This results in the restored trajectory volume being smaller than the actual remaining volume of the pickled mustard tuber after cutting, causing negative residual rates for fibrous skin with a longitudinal stem length of 15 cm and vegetable shape indexes of 1.10, 1.15, and 1.30. During the simulation process, it was found that the stiffness coefficient of spring B has a significant impact on the meat loss rate and fibrous skin residual rate, while the damping of the spring has a relatively small influence on the simulation. Due to the simulation being conducted under certain assumptions, there will inevitably be errors compared to the actual cutting process. However, within a certain range of error, the designed virtual prototype has a certain degree of credibility.

## 6. Conclusions and Future Work

This article presents a metamorphic-constrained, underactuated humanoid pickling machine for peeling pickled mustard tubers. The machine uses a fibrous skin measuring mechanism to obtain the distribution of the fibrous skin of the pickled mustard tubers. The conveyed mechanism transports the pickled mustard tubers, which are then subjected to flexible peeling and fiber removal by the fibrous skin measuring mechanism, peeling mechanism, and return mechanism. Statistical analysis of the morphological characteristics of the pickled mustard tubers was performed, and three basic categories were selected. The arc-shaped surface contour of the pickled mustard tubers was extracted, and three interchangeable tool types with connecting rod lengths of 210 mm, 165 mm, and 125 mm were designed to greatly reduce the loss rate. The support and reaction forces acting on the modular rotation pair and the driving torque and constraint torque acting on the variable-cell rotation pair of each of the three configurations were analyzed. The theoretical feasibility and effectiveness of the proposed metamorphic-constrained underactuated humanoid pickling machine were validated. An ADAMS virtual prototype model was established, and the tool end trajectory simulation and tool metamorphic-constrained optimization for pickling and peeling the mustard tubers were performed. The results showed that the peeling loss rate for each type of pickled mustard tuber was not more than 15%, with an average residual rate of only around 2%. The theoretical peeling rate could reach 15/min, and the optimal stiffness coefficients and damping values for springs *A*, *B*, *C*, *D*, and E were obtained for the tool end trajectory simulation.

In the future, the metamorphic-constrained theory will be further demonstrated, and a working prototype will be manufactured to conduct experiments on peeling pickled mustard tubers. The prototype’s automatic feeding and discharging, fibrous skin sensing module, tool cutting module, and related performance indicators such as loss rate and residual rate will be continuously optimized and improved during the experimental study. Furthermore, the pickled mustard tubers’ 3D point clouds will be projected onto a 2D plane using the spatial structure circular description operator SSCD [32], and the deep histogram will be used to classify the features of the pickled mustard tubers’ 2D plane images. Based on the classification results, tool types that match the pickled mustard tubers will be designed to improve tool–pickled mustard tuber matching and reduce the loss and residual rates, achieving the fully automatic deep pickling and processing of pickled mustard tubers.

## 7. Patents

Lin Zhang et al., Sensorless peeling robot based on variable configuration constraints for pickled mustard tubers, Dutch patent, 202111215022.6, Yangtze Normal University, 2023.5.17 (Granted).

## Figures and Tables

**Figure 1 biomimetics-08-00566-f001:**
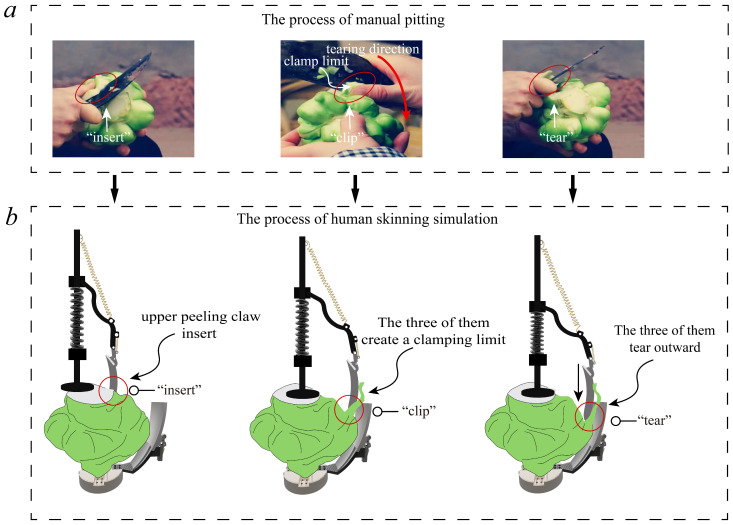
“Insertion-clip-tearing” anthropomorphic skinning process. (**a**) manual peeling process; (**b**) machine-simulated human peeling process.

**Figure 2 biomimetics-08-00566-f002:**
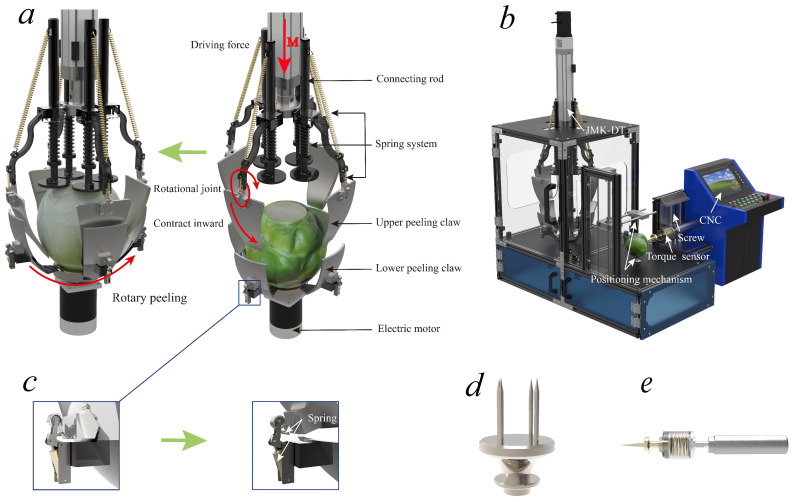
Structural diagram of pickled vegetable de-sinew and peeling device. (**a**) Description of the structure of the muscle peeling device; (**b**) Main structure of the muscle peeling device; (**c**) reset spring; (**d**) Positioning mechanism; (**e**) Torque sensor.

**Figure 3 biomimetics-08-00566-f003:**
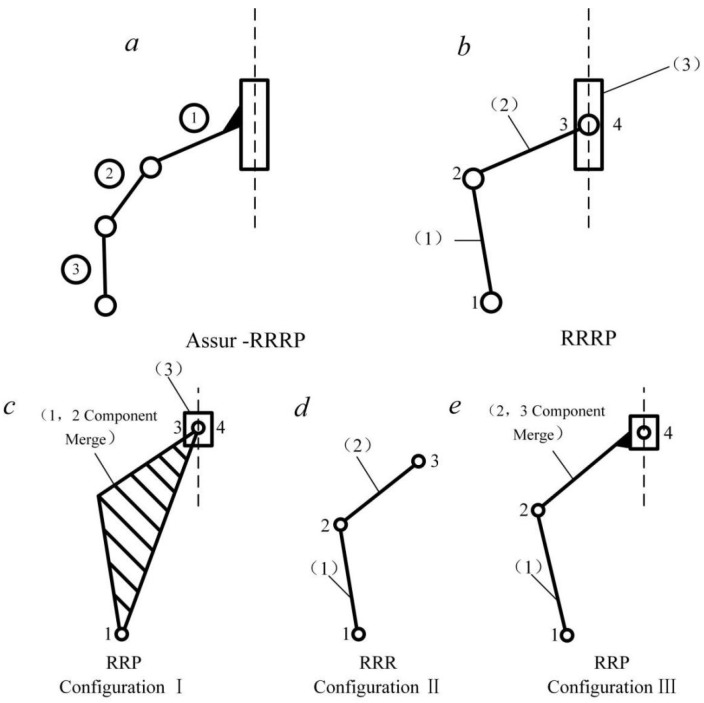
RRRP bar group configuration transformation process. (**a**) Assur-RRRP rod assembly; (**b**) RRRP rod assembly; (**c**) Configuration I; (**d**) Configuration II; (**e**) Configuration III.

**Figure 4 biomimetics-08-00566-f004:**
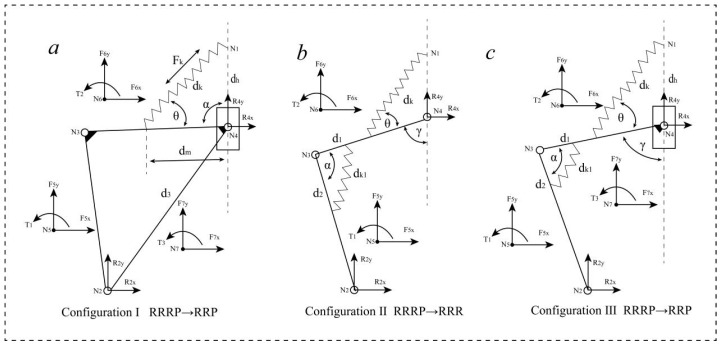
Three kinds of variable configuration metamorphic cell mechanism modular force analysis. (**a**) Configuration I module force analysis; (**b**) Configuration II module force analysis; (**c**) Configuration III module force analysis.

**Figure 5 biomimetics-08-00566-f005:**
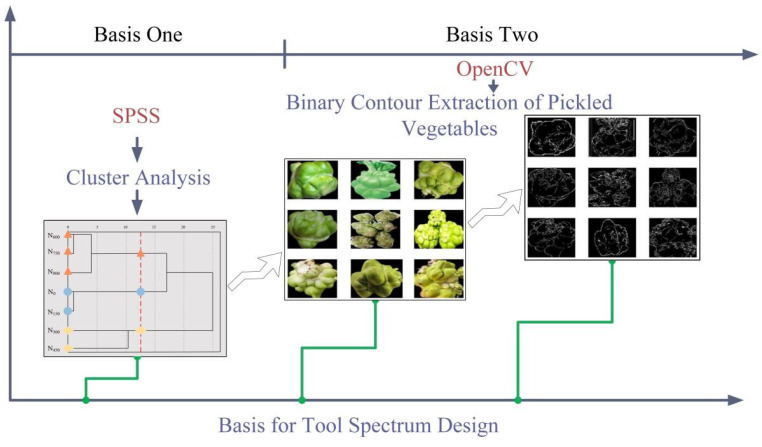
Robot system workflow.

**Figure 6 biomimetics-08-00566-f006:**
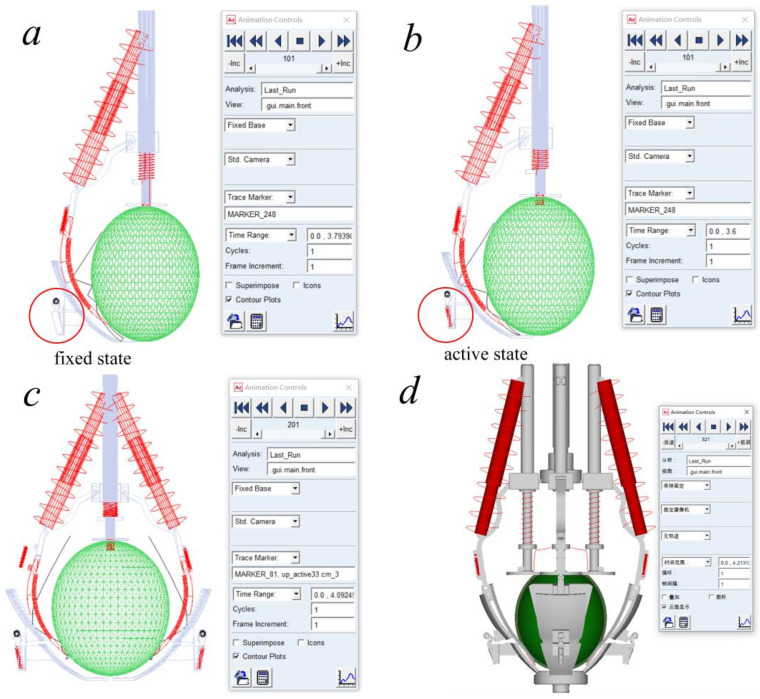
Four different cutting methods. (**a**) Single claw and single blade cutting method with bottom peeling claw fixed; (**b**) Normal single claw and single blade cutting method; (**c**) Single-arm multi-blade cutting method; (**d**) Multi-arm multi-blade cutting method.

**Figure 7 biomimetics-08-00566-f007:**
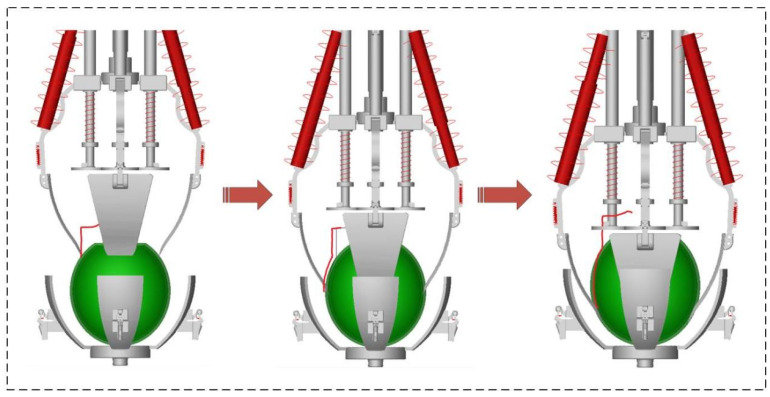
Cutting path diagram.

**Figure 8 biomimetics-08-00566-f008:**
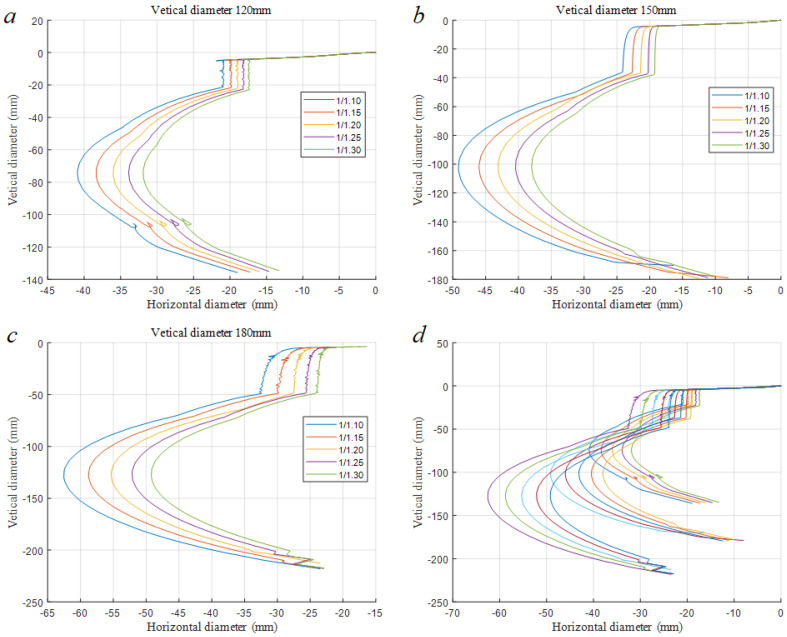
The cutting trace graph. (**a**) The end effector trajectory diagram with a vertical diameter of 120 mm; (**b**) The end effector trajectory diagram with a vertical diameter of 150 mm; (**c**) The end effector trajectory diagram with a vertical diameter of 180 mm; (**d**) End Effector Trajectory Synthesis Diagram.

**Table 1 biomimetics-08-00566-t001:** Metamorphic peeling finger mechanism: symbols and definitions.

Symbol	Definition
*N*_5_, *N*_6_, *N*_7_	Link *N*_2_*N*_3_, *N*_3_*N*_4_, and the centroid of the slider.
*α*	Geometric constraint angle of variable cell revolute pair.
*d_k_*, *d_k_*_1_	Current length of springs 1 and 2
*F_k_*, *F_k_*_1_	Spring force of springs 1 and 2
*θ*	Angle between the force direction of spring 1 and the link *N*_3_*N*_4_.
*d_h_*, *d_m_*	Distance between the two ends of spring 1 and point *N*_4_.
*d*_1_, *d*_2_	Distance between the two ends of spring 2 and point *N*_3_.
*F*_7_, *T*_3_	External force and torque applied to the slider.
*F*_5_, *F*_6_	The link *N*_2_*N*_3_ and *N*_3_*N*_4_ experience forces due to the sliding of the slider.
*T*_1_, *T*_2_	The link *N*_2_*N*_3_ and *N*_3_*N*_4_ experience torques due to the sliding of the slider.
*k*	Spring constant
*a*	Original length of the spring
*P_ijx_*, *P_ijy_*	Distance in the *x* and *y* directions between points *i* and *j*.
*R_ix_*, *R_iy_*	Supporting reaction forces in the *x* and *y* directions acting on motion pair *i*.
*F_ix_*, *F_iy_*	Forces acting on point *i* in the *x* and *y* directions.
*T_ij_*	Force applied at point *i*, causing torque at point *j*.
*Ti*	Torque acting at point *i*.
*T_k_*	The torque due to the spring force at point *N*_4_, i.e., the constraint resistance torque provided by the variable cell subsystem.
*T*_54_, *T*_64_	The torque caused by the gravitational forces of the two links at point *N*_4_.
Δ*T*	The driving torque of the variable cell revolute pair.
*γ*	The angle between the guide and the link *N*_3_*N*_4_.

**Table 2 biomimetics-08-00566-t002:** The influence of nitrogen application rate on quality-related indicators of pickled mustard tuber.

Designation	Plant Height/cm	Plant Width/cm	Shape Index (Longitudinal Diameter/Transverse Stem)	Skin Elasticity/%	Net Yield/%
*N* _0_	45.5	47.9	1.32	7.62	39.39
*N* _150_	46.3	48.4	1.38	8.62	39.89
*N* _300_	46.9	46.2	1.20	8.28	43.35
*N* _450_	44.9	46.9	1.18	12.19	42.33
*N* _600_	43.2	44.3	1.15	7.45	40.42
*N* _750_	44.3	45.3	1.17	7.62	39.86
*N* _900_	44.3	48.0	1.15	6.99	39.03

**Table 3 biomimetics-08-00566-t003:** Label-to-name correspondence table.

**Component Name**	**Component Number**	**Adams Component Name**	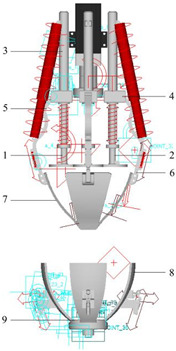
Compression rod	1,2	up_mid	
Upper rod	3	up_mid2	
Block	4	up_active1	
Long link	5	up_active2	
Short link, Upper gripper	6,7	up_active3	
Lower gripper	8	under_active	
Placement board	9	under_stop1	
		under_stop2	
		under_stop3	

**Table 4 biomimetics-08-00566-t004:** The simulation parameters of the spring system when the pickled mustard tuber’s longitudinal stem is 12 cm.

volume of pickled mustard tubers	vertical diameter/ horizontal diameter	1.10	1.15	1.20	1.25	1.30
	horizontal diameter	11.82	11.30	10.83	10.40	10.00
component name	parameter type	simulation-optimized parameter results				
Spring a	stiffness coefficient	8 × 10^2^	8 × 10^2^	8 × 10^2^	8 × 10^2^	8 × 10^2^
	damping coefficient	5 × 10^2^	5 × 10^2^	5 × 10^2^	5 × 10^2^	5 × 10^2^
Spring b	stiffness coefficient	1.6 × 10^4^	1.7 × 10^4^	1.8 × 10^4^	1.9 × 10^4^	2 × 10^4^
	damping coefficient	1 × 10^3^	1 × 10^3^	1 × 10^3^	1 × 10^3^	1 × 10^3^
	Pre-tightening Force	2.5 × 10^1^	2.5 × 10^1^	2.5 × 10^1^	2.5 × 10^1^	2.5 × 10^1^
Spring c	stiffness coefficient	1 × 10^4^	1 × 10^4^	1 × 10^4^	1 × 10^4^	1 × 10^4^
	damping coefficient	1 × 10^3^	1 × 10^3^	1 × 10^3^	1 × 10^3^	1 × 10^3^
Spring d	stiffness coefficient	1 × 10^4^	1 × 10^4^	1 × 10^4^	1 × 10^4^	1 × 10^4^
	damping coefficient	8 × 10^2^	8 × 10^2^	8 × 10^2^	8 × 10^2^	8 × 10^2^
Spring e	stiffness coefficient	1 × 10^4^	1 × 10^4^	1 × 10^4^	1 × 10^4^	1 × 10^4^
	damping coefficient	8 × 10^2^	8 × 10^2^	8 × 10^2^	8 × 10^2^	8 × 10^2^

**Table 5 biomimetics-08-00566-t005:** The simulation parameters of the spring system when the pickled mustard tuber’s longitudinal stem is 15 cm.

volume of pickled mustard tubers	vertical diameter/ horizontal diameter	1.10	1.15	1.20	1.25	1.30
	horizontal diameter	13.64	13.04	12.50	12.00	11.54
component name	parameter type	simulation-optimized parameter results				
Spring a	stiffness coefficient	8 × 10^2^	8 × 10^2^	8 × 10^2^	8 × 10^2^	8 × 10^2^
	damping coefficient	5 × 10^2^	5 × 10^2^	5 × 10^2^	5 × 10^2^	5 × 10^2^
Spring b	stiffness coefficient	1.2 × 10^4^	1.3 × 10^4^	1.4 × 10^4^	1.5 × 10^4^	1.6 × 10^4^
	damping coefficient	6 × 10^3^	6 × 10^3^	6 × 10^3^	6 × 10^3^	6 × 10^3^
	Pre-tightening Force	2.5 × 10^1^	2.5 × 10^1^	2.5 × 10^1^	2.5 × 10^1^	2.5 × 10^1^
Spring c	stiffness coefficient	1 × 10^4^	1 × 10^4^	1 × 10^4^	1 × 10^4^	1 × 10^4^
	damping coefficient	1 × 10^3^	1 × 10^3^	1 × 10^3^	1 × 10^3^	1 × 10^3^
Spring d	stiffness coefficient	1 × 10^4^	1 × 10^4^	1 × 10^4^	1 × 10^4^	1 × 10^4^
	damping coefficient	8 × 10^2^	8 × 10^2^	8 × 10^2^	8 × 10^2^	8 × 10^2^
Spring e	stiffness coefficient	1 × 10^4^	1 × 10^4^	1 × 10^4^	1 × 10^4^	1 × 10^4^
	damping coefficient	8 × 10^2^	8 × 10^2^	8 × 10^2^	8 × 10^2^	8 × 10^2^

**Table 6 biomimetics-08-00566-t006:** The simulation parameters of the spring system when the pickled mustard tuber’s longitudinal stem is 18 cm.

volume of pickled mustard tubers	vertical diameter/ horizontal diameter	1.10	1.15	1.20	1.25	1.30
	horizontal diameter	16.36	15.65	15.00	14.40	13.85
component name	parameter type	simulation-optimized parameter results				
Spring a	stiffness coefficient	6 × 10^3^	6 × 10^3^	6 × 10^3^	6 × 10^3^	6 × 10^3^
	damping coefficient	5 × 10^2^	5 × 10^2^	5 × 10^2^	5 × 10^2^	5 × 10^2^
Spring b	stiffness coefficient	8 × 10^3^	9 × 10^3^	1 × 10^4^	1.1 × 10^4^	1.2 × 10^4^
	damping coefficient	6 × 10^3^	6 × 10^3^	6 × 10^3^	6 × 10^3^	6 × 10^3^
	Pre-tightening Force	2.5 × 10^1^	2.5 × 10^1^	2.5 × 10^1^	2.5 × 10^1^	2.5 × 10^1^
Spring c	stiffness coefficient	1 × 10^4^	1 × 10^4^	1 × 10^4^	1 × 10^4^	1 × 10^4^
	damping coefficient	1 × 10^3^	1 × 10^3^	1 × 10^3^	1 × 10^3^	1 × 10^3^
Spring d	stiffness coefficient	1 × 10^4^	1 × 10^4^	1 × 10^4^	1 × 10^4^	1 × 10^4^
	damping coefficient	8 × 10^2^	8 × 10^2^	8 × 10^2^	8 × 10^2^	8 × 10^2^
Spring e	stiffness coefficient	1 × 10^4^	1 × 10^4^	1 × 10^4^	1 × 10^4^	1 × 10^4^
	damping coefficient	8 × 10^2^	8 × 10^2^	8 × 10^2^	8 × 10^2^	8 × 10^2^

**Table 7 biomimetics-08-00566-t007:** Meat loss and residual fibrous tissue rate table.

Vertical Diameter of Pickled Mustard Tubers	Vegetable Shape Index	Total Volume/cm^3^	Skin Volume/cm^3^	Trajectory Reconstruction Volume/cm^3^	Meat Loss Rate%	Skin Tendon Residue Rate%
18 cm	1.10	2898.91	234.99	2666.50	8.02	1.10
	1.15	2652.32	221.60	2441.11	7.96	4.69
	1.20	2435.90	209.58	2240.45	8.02	6.74
	1.25	2244.93	198.72	2064.75	8.03	9.33
	1.30	2075.58	188.87	1903.54	8.29	8.91
15 cm	1.10	1731.15	165.94	1550.29	10.45	−8.99
	1.15	1583.89	156.45	1416.34	10.58	−7.09
	1.20	1454.65	147.93	1309.40	9.99	1.81
	1.25	1340.61	140.24	1201.79	10.36	1.01
	1.30	1239.47	133.27	1104.62	10.88	−1.18
12 cm	1.10	928.55	108.86	823.31	11.33	3.33
	1.15	849.56	102.61	752.63	11.41	5.54
	1.20	780.24	97.00	684.22	12.31	1.01
	1.25	719.07	91.93	628.75	12.56	1.75
	1.30	664.83	87.33	579.29	12.87	2.05

## Data Availability

Data are contained within the article.

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
