# Peer review of "Underactuated Humanoid Peeling Approach for Pickled Mustard Tuber Based on Metamorphic Constraints"

_biomimetics, 2023, doi:10.3390/biomimetics8080566_

Round 1

Reviewer 1 Report

Comments and Suggestions for Authors

1.      This paper extensively analyzes and simulates the pickled mustard tuber peeling robot, revealing that the meat loss rate remains below 15%, achieving a theoretical peeling rate of 15 pieces per minute. Furthermore, the average residual rate of old fibers is approximately 2%. These findings establish a theoretical groundwork for fully automated mustard peeling, complemented by a patented concept. The design and analytical outcomes presented in this paper offer valuable insights for the future development of mustard peeling machines.

2.      The explanation of the meaning of RRRP, RRR, and RRP in Section 3 should be relocated to Section 2.2, where these abbreviations are initially introduced. References should be cited following the journal format, such as [1], there are many incorrectly formatted of references in this paper. Implementing these revisions will enhance readability for the readers.

3.      What is the relevance of reference 31 to this study? (In order to optimize the simulation results and improve the stability of the mechanism, the peeling mechanism is divided into three size specifications [31], each corresponding to the respective longitudinal diameter of the pickled mustard tuber.)

Reviewer 2 Report

Comments and Suggestions for Authors
  • The manuscript deals with the study of an innovative robot for peeling mustard tubers. The subject is very interesting and original and it belongs to the branch of applied research that, in my opinion, is very meaningful.
  • Paper appears clear, relevant for the field and presented in a well-structured manner. 
  • Bibliography is good, it appears complete and does not include an excessive number of self-citations
  • The research appears, at the moment, without experimental tests. The described study deals with the simulation and nominal properties of the device.  
  • The manuscript’s results appear reproducible based on the details given in the methods section. However, this part of the paper must be improved. Even if used symbols are actually described in the text, it is very difficult to interpret them and verify formulas from (1) to (11) having as only reference Figure 4. I suggest to build a table where all the variables and parameters refer to a bigger figure and are described "at a glance" without continuously referring to the text. Please verify formulas (2).
  • Conclusions appear consistent. It is obvious that only the experimental tests will complete this research that, at the moment, describes only the simulation results.
  • Ethics statements and data availability statements are adequate.

Round 2

Reviewer 2 Report

Comments and Suggestions for Authors

Very good improvement of the paper. Formulas appear clear and parameters have been well described.